# Minimal Infiltrative Disease Identification in Cryopreserved Ovarian Tissue of Girls with Cancer for Future Use: A Systematic Review

**DOI:** 10.3390/cancers15174199

**Published:** 2023-08-22

**Authors:** Monika Grubliauskaite, M. E. Madeleine van der Perk, Annelies M. E. Bos, Annelot J. M. Meijer, Zivile Gudleviciene, Marry M. van den Heuvel-Eibrink, Jelena Rascon

**Affiliations:** 1Center for Pediatric Oncology and Hematology, Vilnius University Hospital Santaros Klinikos, Santariskiu Str. 4, LT-08406 Vilnius, Lithuania; 2Life Sciences Center, Vilnius University, Sauletekio Ave. 7, LT-10257 Vilnius, Lithuania; 3Department of Biobank, National Cancer Institute, Santariskiu Str. 1, LT-08406 Vilnius, Lithuania; 4Princess Máxima Center for Pediatric Oncology, 3584 CS Utrecht, The Netherlands; 5Department of Reproductive Medicine, University Medical Center Utrecht, 3584 CX Utrecht, The Netherlands; 6Faculty of Medicine, Vilnius University, M. K. Ciurlionio Str. 21/27, LT-03101 Vilnius, Lithuania; 7Division of Child Health, UMCU-Wilhelmina Children’s Hospital, 3584 EA Utrecht, The Netherlands

**Keywords:** fertility restoration, ovarian tissue, cryopreservation, cancer survivors, pediatric patients, minimal infiltrative disease, minimally residual disease

## Abstract

**Simple Summary:**

Cryopreservation of ovarian tissue and transplantation are the only available fertility techniques to preserve fertility and endocrine function for prepubertal girls with cancer who require immediate cancer treatment. The primary objective of this systematic review was to identify and critically appraise the existing evidence regarding targeted minimal infiltrative disease detection in harvested ovarian tissues and identify markers that may be of value for assessment before autotransplantation, thereby facilitating fertility restoration in childhood cancer survivors. While the majority of malignancies were found to be at low risk of containing malignant cells in ovarian tissue, more studies are needed to ensure safe implementation of future fertility restoration in clinical practice.

**Abstract:**

Background: Ovarian tissue cryopreservation and transplantation are the only available fertility techniques for prepubertal girls with cancer. Though autotransplantation carries a risk of reintroducing malignant cells, it can be avoided by identifying minimal infiltrative disease (MID) within ovarian tissue. Methods: A broad search for peer-reviewed articles in the PubMed database was conducted in accordance with PRISMA guidelines up to March 2023. Search terms included ‘minimal residual disease’, ‘cryopreservation’, ‘ovarian’, ‘cancer’ and synonyms. Results: Out of 542 identified records, 17 were included. Ovarian tissues of at least 115 girls were evaluated and categorized as: hematological malignancies (*n* = 56; 48.7%), solid tumors (*n* = 42; 36.5%) and tumors of the central nervous system (*n* = 17; 14.8%). In ovarian tissue of 25 patients (21.7%), MID was detected using RT-qPCR, FISH or multicolor flow cytometry: 16 of them (64%) being ALL (*IgH* rearrangements with/without *TRG*, *BCL-ABL1*, *EA2-PBX1*, *TEL-AML1* fusion transcripts), 3 (12%) Ewing sarcoma (*EWS-FLI1* fusion transcript, *EWSR1* rearrangements), 3 (12%) CML (*BCR-ABL1* fusion transcript, *FLT3*) and 3 (12%) AML (leukemia-associated immunophenotypes, *BCR-ABL1* fusion transcript) patients. Conclusion: While the majority of malignancies were found to have a low risk of containing malignant cells in ovarian tissue, further studies are needed to ensure safe implementation of future fertility restoration in clinical practice.

## 1. Introduction

Substantial improvements in childhood cancer treatment, as well as enhanced stratification and refinement of supportive care, have resulted in a five-year survival rate of over 80% in high-income countries [1]. Hence, substantial growth in the childhood cancer survivor population is currently observed, which urges attention toward treatment-related late sequelae. Impaired fertility is one of the relevant long-term adverse effects of pediatric cancer treatment, which is mainly caused by gonadotoxic chemotherapy and/or radiotherapy. Alkylating agents are especially notorious for gonadotoxicity and can lead to delayed puberty, premature menopause, and ovarian insufficiency due to a substantial decrease in the follicle pool [2]. Nowadays, fertility issues among young cancer survivors are gaining increasing attention. The involvement of a multidisciplinary fertility specialist team is crucial to ensure a timely assessment of the gonadal damage risk and a proper fertility preservation decision process to improve the quality of life in remission [3].

Cryopreservation of ovarian tissue (OT) is one of the options to preserve fertility in prepubertal and adolescent female patients who require immediate cancer treatment [3]. To preserve follicles, ovarian tissue cryopreservation (OTC) is the only option since ovarian stimulation cannot be performed. OTC provides the opportunity to store ovarian tissue and to autotransplant it once the patient is disease-free and has a child-wish. According to recent data, 1019 OTC procedures have been performed in children and young adults with cancer (0–20.4 years), of whom 298 were younger than 13 years. Eighteen of them received an ovarian tissue (OT) transplant in adulthood, resulting in eleven pregnancies, of which nine resulted in a live birth [4,5]. In addition, hormonal function restoration was observed in 17 (94%) patients who had undergone OT transplantation [5]. In 2018, it was shown that 95% of 318 women who had OT transplantation worldwide had successful long-term restoration of ovarian and endocrine functions as adults [6]. Currently, alternatives to OT transplantation are being developed, such as in vitro activation and/or ex vivo growth and maturation of primordial follicles [7,8], construction of artificial ovaries [9] or developing oocytes from stem cells [10,11]. Unfortunately, the biological principles of follicular development are still not fully understood [8]. To give rise to high-quality oocytes, OT freezing should ideally be performed before treatment [12]. 

An important concern of OTC is the risk of reintroducing malignant cells that may be harbored within the OT [13,14,15]. Methods of identification of minimal infiltrative disease (MID) in cryopreserved OT are of crucial importance to prevent iatrogenic transplantation of cancer cells and subsequent relapse [6]. MID in cryopreserved OT can be identified using (a combination of) histology, immunohistochemistry (IHC) and currently available molecular technologies. Subsequently, purging strategies for residual cells in intact OT are being developed [16,17,18,19]. All of which aim to eliminate the risk of autotransplanting residual cancer cells. 

This systematic review was pursued as part of The Twinning in Research and Education to Improve Survival in Childhood Solid Tumours in Lithuania (TREL) project, which is an EU Horizon 2020-funded project that aims to improve various aspects of childhood cancer care, including quality of survivorship, by the twinning of researchers from different institutions. We aimed to answer the following questions:Which targets are currently being used to screen for MID in cryopreserved OTs of pediatric patients?Which techniques were used for MID detection in OT in the relevant subgroups and may be used for assessment of the graft before autotransplantation?

## 2. Materials and Methods

This systematic review was conducted in accordance with the Preferred Reporting Items for Systematic Reviews and Meta-Analyses (PRISMA) guidelines (www.prisma-statement.org/ (accessed on 18 July 2023)).

### 2.1. Search Strategy

Using the PubMed database, a broad search for peer-reviewed articles was carried out in February 2022 and updated on 7 March 2023. In this search, terms including ‘minimal residual disease’, ‘cryopreservation’, ‘ovarian’ and ‘cancer’ and their synonyms were used (Table A1). Papers addressing the presence or absence of malignant cells in human OT harvested for cryopreservation were included, as were all papers describing human–animal and human–human ovarian transplantation after malignant disease. The selection and filtering of relevant articles were performed by two independent authors (M.G. and M.E.M.v.d.P.) using the systematic review web application Rayyan (http://rayyan.qcri.org (accessed on 6 February 2022)) [20] (Figure 1). The reference list of identified articles was also screened to identify additional important articles.

### 2.2. Selection Criteria

For both aims, the inclusion and exclusion criteria can be found in Table A2. Briefly, all articles in English with fully available text on cryopreserved OT of female childhood (<18 years) cancer patients which included assessments of potential biomarkers for cancer cells in OT were included in this systematic review.

Excluded articles described adults or pediatric male patients, research solely including cancer cell lines or markers measured only in the origin tissue of the cancer, blood or serum, reviews, comments and guidelines.

### 2.3. Data Extraction

The information extracted from every identified article included the author and year of publication, study type, number and characteristics of patients (cancer diagnosis, age at the cryopreservation, received cancer treatment before OTC), MID markers and methods used to detect them and outcomes—presence/absence of MID.

### 2.4. Quality of Evidence Assessment

Two authors (M.G. and M.E.M.v.d.P.) evaluated the methodological quality of the included studies using the Quality in Prognosis Studies (QUIPS) tool criteria independently [22] and evaluated the overall quality of evidence in every study using the GRADE criteria (Appendix A) [23]. The following six domains of the QUIPS tool were used: (1) study participation; (2) study attrition; (3) prognostic factor measurement; (4) outcome measurement; (5) study confounding; and (6) statistical analysis and reporting. Each domain was assessed as having a low, moderate or high risk of bias and subsequently an overall risk of bias was established. The most important domains for our systematic review were study participation, prognostic factor measurement, outcome measurement and study confounding since we aimed to evaluate which markers and methods can be used to detect MID in OT. The most important influential confounders for detection of MID included cancer treatment before OTC, no availability of original tumor material or markers and survival or disease recurrence. Disagreement between the authors was resolved by consensus.

## 3. Results

A total of 515 hits were identified in the PubMed database and 14 additional articles were included from studies’ references or were recommended (Figure 1). The updated search on 7 March 2023 revealed 27 new hits. After the removal of duplicates and screening of titles and abstracts, 358 articles were excluded. Based on title–abstract screening, a full-text screen was performed on 195 articles and 17 articles dated from 2010–2022 were included in the qualitative synthesis.

The studies were grouped into three categories: (1) solid tumors including EWS, rhabdomyosarcoma, osteosarcoma, synovial sarcoma and clear cell sarcoma (*n* = 7); (2) hematological malignancies including acute lymphoblastic leukemia (ALL), acute myeloid leukemia (AML), chronic myeloid leukemia (CML), juvenile myelomonocytic leukemia (JMML) and Burkitt’s lymphoma (*n* = 8); (3) tumors of the central nervous system (CNS) and brain tumors including astrocytoma, ependymoma, germinoma, glioblastoma, medulloblastoma and primitive neuroectodermal tumors (*n* = 2). No studies describing patients with Wilms tumors, hepatoblastomas, retinoblastomas or Hodgkin lymphomas were identified. Evaluation of the articles is shown in Table 1, Table 2, Table 3 and Appendix A. The markers and methods described in the included articles are presented in Table 1, Table 2 and Table 3. In this systematic review, ovarian tissue from 115 to 122 girls (some studies recorded the age of patients in a range that included patients older than 18 years) was reported to be analyzed by histology, IHC, several types of polymerase chain reaction (PCR), fluorescence in situ hybridization (FISH) or multicolor flow cytometry (MFC). For 62 of the patients, additionally, OT was xenotransplanted in immunodeficient mice and evaluated for MID using histology, IHC, PCR or MFC methods [16,24,25,26,27,28,29]. The main cryopreservation technique used was slow freezing (used for 87 patients, 5 of them had additionally snap frozen OT pieces, while for the others the used technique was not reported). The risk of bias in most studies was moderate to high and was based primarily on study participation, inconsequent use of the methods to detect MID and insufficiently taking possible confounders into account. The overall quality of evidence was very low, primarily due to the study type (case reports and retrospective studies) and low number of participants.

**Table 1 cancers-15-04199-t001:** Summary studies on minimal infiltrative disease in cryopreserved ovarian tissue of girls with solid tumors.

Ref.	Study Type	Pt (*n*)	Type of Cancer	Markers and Methods (1–5)	Detected by Methods(1–5)	Positive Patients, *n* (%)	Potential Bias	Overall Bias
Histology (1)	IHC (2)	FISH (3)	PCR ^$^ (4)	Xeno (5)	SB	AB	MB	DB	SC	SA*	
[30]	Retrospective study	7	Ewing sarcoma	*n* = 6	CD99*n* = 6	-	*EWS-FLI 1* FT*n* = 5	-	4	method4: 1/5 (20%)							
[31]	Retrospective study	5	Ewing sarcoma	*n* = 5	-	*EWSR1 n* = 5	*EWS-FLI 1* FT*n* = 5	-	ND	-							
[32]	Case report ^‡^	1	Ewing sarcoma	-	-	-	*EWS-FLI 1* FT*n* = 1	-	ND	-							
[15]	Case reports	2	Ewing sarcoma	-	CD99*n* = 2	*EWSR1 n* = 2	-	-	2, 3	method2: 2/2 (100%)method3: 2/2 (100%)							
[24]	Retrospective study	9	Ewing sarcoma, synovial sarcoma, osteosarcoma	*n* = 9	-	-	*EWS-FLI 1* FT*n* = 6	*n* = 9	ND	-							
[33]	Retrospective study	2–9 ^†^	Ewing sarcoma, osteosarcoma	*n* = 5	CD99*n* = 2	-	*EWS-ERG* FT*n* = 2	-	ND	-							
[34]	Retrospective study	16	Ewing sarcoma,PNET, clear cellsarcoma, synovial sarcoma,rhabdomyosarcoma	*n* = 16	CD99, MDM2, myogenin, S100, MyoD 1, melanoma cocktail*n* = 16	-	*EWS-FLI 1*, *EWS-ERG*, *PAX3-FOXO1*, *PAX7-FOXO1*, *SYT-SSX*, *EWS-ATF1* FTs, *MyoD1* *n* = 12	-	ND	-							
**Grade assessment**
Study design	Low	Retrospective cohort studies in 5/7; case reports in 2/7
Study limitations	−2	Important limitations: SB high in 7/7; AB low in 5/7, moderate in 1/7, high in 1/7; MB is low in 3/7, moderate in 3/7, high in 1/7; DB is low in 3/7, moderate in 3/7, high in 1/7; SC is low in 5/7, moderate in 2/7; SA is low in 4/7, moderate in 2/7, high in 1/7
Consistency	0	Little inconsistency: 6/7 (1/6 studies detected MID) studies used PCR and 4/7 IHC (1/4 studies detected MID) to detect MID; FISH detected MID in 1/2, and markers used were mostly the same across all the studies for a specific method)
Directness	−1	Though results are direct, population and outcomes cannot be generalized
Precision	−1	Low number of patients
Publication bias	0	Unlikely
Effect size	0	NA
Dose–response	0	NA
Plausible confounding	+1	Treatment as plausible confounder may seriously influence the effect
**Quality of evidence**	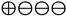	**Very low**
**Conclusion**		Detection of MID in ovarian tissues from solid tumor patients is unlikely despite various attempts to identify it but the results cannot be generalized (7 studies, 42–49 ^†^ participants)


 = low risk of bias; 

 = moderate risk of bias; 

 = high risk of bias; 
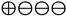
 = very low quality of evidence. AB = attrition bias (the study data available (i.e., participants not lost to follow up) adequately represent the study sample), DB = detection bias (the outcome of interest is measured in a similar way for all participants), FISH = fluorescence in situ hybridization, FT = fusion transcript, IHC = immunohistochemistry, L = low risk, M = moderate risk, MB = measurement bias (the prognostic factor is measured in a similar way for all participants), ND = not detected, OT = ovarian tissue, OTC = ovarian tissue cryopreservation, PCR = polymerase chain reaction, PNET = primitive neuroectodermal tumor, Pt = patients, SA* = no statistical analysis performed, all primary outcomes are reported, SB = selection bias (the study sample adequately represents the population of interest), SC = study confounding (important potential confounding factors are appropriately accounted for), Xeno = xenotransplantation (and evaluation after). ‘^$^’ includes all types of PCR: PCR, RT-PCR, quantitative RT-PCR, digital droplet RT-PCR, nested PCR; ‘-’ means the method was not used in the specific study; ^†^ OS: *n* = 7 (mean age 15, range 13–18 y) EW: *n* = 4 (mean 19, range 17–21 y), it is unclear how many patients were <18 years; ^‡^ letter to the editor about a case report.

### 3.1. Solid Tumors

Seven studies investigated MID in different solid tumors such as Ewing sarcoma, primitive neuroectodermal tumor (PNET), osteosarcoma, rhabdomyosarcoma, synovial sarcoma and clear cell sarcoma. In total, the OT of at least 42 patients with a mean age at the time of cryopreservation of 12.2 ± 3.6 years (median 13 years, range 1–17 years) was analyzed. Eleven patients had metastatic disease, twenty-four had localized disease and seven patients did not have metastasis status reported. Thirty patients had their OT frozen using slow freezing technique, five additionally had their OT snap frozen, while for twelve patients the used technique was not specified. EWS was the most common malignancy (27 out of 42 patients). The primary tumor or blood was used as a positive control to find MID markers in the OT, if possible. Histology, IHC, reverse transcription quantitative PCR (RT-qPCR), FISH and xenotransplantation were used for MID detection (Table 1). In summary, in 3 OT samples from 27 (11%) EWS patients, an infiltration of cancer cells was found while infiltrations were absent in the OT samples of patients with other solid tumors (*n* = 15). Twenty-nine patients had OTC performed before treatment initiation, ten after treatment initiation and for three patients it was not reported.

***Ewing sarcoma***—OTs of at least 27 pediatric EWS patients were analyzed and in 3 patients cancer cells were found by immunostaining or molecular testing (Table 1). Two out of three MID-positive patients did not have distant metastasis at the time of OTC reported [15] and for the third one the disease spread was not mentioned [30]. Seven of the twenty-four MID-negative patients had metastatic disease, for six patients it was not described and eleven patients had localized disease. Positive identification of MID included CD99-positive tumor islets, *EWS-FLI1* fusion transcript and *EWSR1* rearrangements detected by IHC, RT-qPCR and FISH analysis, accordingly. Histological evaluation or xenotransplantation did not detect infiltration of malignant cells. Interestingly, in peripheral blood of one EWS patient the result was positive for fusion transcript though OT evaluation showed the absence of MID [30].

***PNET (Ewing family)***—OTs of three patients with localized PNET of the Ewing family were found to be negative for MID [34]. OTs were analyzed by histology, IHC or RT-qPCR using CD99 and NSE or *EWS-FLI1*, accordingly.

***Osteosarcoma***—In OTs from osteosarcoma (OS) patients (*n* = 2–7 (girls were presented in an age range including patients >18 years; precise number is not known)), only histology was used to check for MID and no malignant cells were found [24,33]. Two patients were reported to have metastatic disease.

***Soft tissue sarcomas***—Malignant cells were absent in all ten patients, though two had metastatic disease (both deceased) and one had recurrence [34]. Detection of MID for rhabdomyosarcoma included myogenins, MyoD1 and *PAX-FOXO1* fusion transcript, for synovial sarcoma—Bcl-2 protein and *SYT-SSX* fusion transcript, for clear cell sarcoma—S100 and melanoma cocktail and *EWS-ATF1* fusion transcript by IHC and RT-qPCR [34]. Histological evaluation after xenotransplantation showed no MID infiltration [24].

### 3.2. Hematological Malignancies

Eight studies performed MID analysis in OT in different hematological malignancies including ALL, AML, CML, JMML and Burkitt’s lymphoma. In total, 56 harvested OTs were analyzed from patients with a mean age at the time of cryopreservation of 10.4 ± 5.0 years (median 13 years, range 1–17 years). All patients except one (not reported) had their OTs frozen using the slow freezing technique, in addition, 5 of those patients had their OT frozen by snap freezing technique as well. Bone marrow or a blood sample from each patient was used as a positive control and analyzed to find specific markers for MID detection, if possible (Table 2). Methods included histology, IHC, RT-PCR, MFC or xenotransplantation into immunodeficient mice. In summary, in 22 out of 56 (39%) analyzed patients, the infiltration of malignant cells was found. The OTC was performed at various time points, 10 patients showed positive MID before chemotherapy and 12 after the initiation of the treatment, while 15 out of 34 MID-negative patients had OTC performed before chemotherapy and 19 after having initiated the treatment.

**Table 2 cancers-15-04199-t002:** Summary studies on minimal infiltrative disease in cryopreserved ovarian tissue of girls with hematological malignancies.

Ref.	Study Type	Pt(*n*)	Type of Cancer	Markers and Methods (1–5)	Detected by Methods (1–5)	Positive Patients, *n* (%)	Potential Bias	Overall Bias
Histology (1)	IHC (2)	PCR ^$^ (3)	Xeno (4)	MFC (5)	SB	AB	MB	DB	SC	SA*	
[35]	Case report	1	CML	*n* = 1	Glycophorin A, MPO, CD34, CD68, LCA/DC45, Factor VIII*n* = 1	*BCR-ABL* FT*n* = 1	-	-	3	1/1 (100%)							
[25]	Retrospective study	8	ALL, AML	*n* = 8	-	*ETV6-RUNX1*, *BCR-ABL1* FTs, *IGH*, *TCR*, *FLT3* rearrangements*n* = 6	*n* = 8	-	1, 3, 4	method1: 1/8 (12.5%)method3: 5/6 (83%)method4: 2/8 (25%)							
[36]	Case report	1	ALL	-	-	*BCR-ABL* FT*n* = 1	-	-	ND	-							
[37]	Retrospective study	14	ALL, AML, CML, JMML	*n* = 14	CD34, CD10, CD20, CD79a, CD3, TdT, CD117, MPO, CD68*n* = 14	*TEL-AML1*, *BCR-ABL b2a2*, *BCR-ABL b2a2*, *CBFB-MYH11* type A FTs*n* = 5	-	-	3	3/5 (60%)							
[26]	Retrospective study	8	B-ALL,T-ALL, AML	-	-	-	*n* = 6	CD19, CD34, CD10-, negative for myeloid markers/CD45, HLA-DR2, CD10, CD19, CD22, CD33/CD19, CD10, CD22, CD38/CD45, CD10, CD19, CD22, CD34, HLA-DR2/CD2, cyCD3, CD5, CD7, CD10, CD33, CD34, CD45RA, CD123*n* = 8	4, 5	method4: 1/6 (12.5%)method5: 2/8 (25%)							
[38]	Retrospective study	9	ALL, AML, Burkitt’s lymphoma	*n* = 9	-	*E2A-PBX1*, *TEL-AML1*, *MLL-AF4*, *AML1-ETO* FTs, *IgK Kde*, *IgH*, *TCRD*, *TCRB**n* = 9	-	-	3	2/9 (22%)							
[27]	Retrospective study	10	ALL, CML	*n* = 10	-	*BCR-ABL1*, *Ig* and/or *TCR-gama* rearrangements*n* = 8	*n* = 10	-	3, 4	method3: 5/8 (62.5%)method4: 4/10 (40%)							
[16]	Retrospective study	5	ALL	*n* = 5	CD20, CD79a, CD3, TdT*n* = 5	*IgH* rearrangements,*EA2-PBX1* FT *n* = 4	*n* = 5	-	3	4/4 (100%)							
**Grade assessment**
Study design	Low	Retrospective cohort studies in 6/8; case reports in 2/8
Study limitations	−1	Some limitations: SB high in 8/8; AB low in 8/8; MB is low in 4/8, moderate in 3/8, high in 1/8; DB is low in 4/8, moderate in 3/8, high in 1/8; SC low in 5/8, moderate in 3/8; SA low in 8/8
Consistency	−1	Though MID was detected by PCR in 6/7 studies and by xenotransplantation in 3/4, IHC detected MID in 0/3 and histology in 1/6 studies. The markers analyzed were very different across the studies
Directness	−1	Though results are direct, population and outcomes cannot be generalized
Precision	−1	Low number of events
Publication bias	0	Unlikely
Effect size	0	NA
Dose–response	0	NA
Plausible confounding	+1	Treatment as plausible confounder may seriously influence the effect
**Quality of evidence**	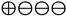	**Very low**
**Conclusion**		Detection of MID in ovarian tissues from the patients of haematological malignancies is moderately likely, despite various attempts to identify it, the results cannot be generalized (8 studies, 56 participants)


 = low risk of bias; 

 = moderate risk of bias; 

 = high risk of bias; 
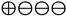
 = very low quality of evidence. AB = attrition bias (the study data available (i.e., participants not lost to follow up) adequately represent the study sample), ALL = acute lymphoblastic leukemia, AML = acute myeloid leukemia, CML = chronic myeloid leukemia, DB = detection bias (the outcome of interest is measured in a similar way for all participants), FISH = fluorescence in situ hybridization, FT = fusion transcript, IHC = immunohistochemistry, JMML = juvenile myelomonocytic leukemia, L = low risk, M = moderate risk, MB = measurement bias (the prognostic factor is measured in a similar way for all participants), MID = minimally infiltrative disease, MPO = myeloperoxidase, ND = not detected, OT = ovarian tissue, OTC = ovarian tissue cryopreservation, PCR = polymerase chain reaction, Pt = patients, SA* = no statistical analysis performed, all primary outcomes are reported, SB = selection bias (the study sample adequately represents the population of interest), SC = study confounding (important potential confounding factors are appropriately accounted for), Xeno = xenotransplantation (and evaluation after). ‘-’ means the method was not used in the specific study; ‘^$^’ includes all types of PCR: PCR, RT-PCR, quantitative RT-PCR, digital droplet RT-PCR, nested PCR.

***ALL***—OTs of 39 patients were analyzed in 7 studies, and in 16 of them (44%) leukemic cells were found (Table 2). Eleven out of sixteen MID-positive patients had started chemotherapy, while others (*n* = 5) received OTC before the start of the treatment. Fifteen patients tested negative for MID after starting chemotherapy, while eight patients before chemotherapy. PCR or RT-PCR were used for targeted immunoglobulin (*Ig*) gene rearrangements alone or in combination with T-cell receptor gamma, *ETV6-RUNX1* [25,27], *EA2-PBX1* [16,38], *BCR-ABL1* [27] or *TEL-AML1* fusion transcript [37,38] detection in MID-positive OTs. In half of those patients (*n* = 8), leukemic cells were detected in OTs prior to and after xenotransplantation as well [16,25,27]. 

While, in some cases, MID was detectable in OT prior to transplantation into mice, recovered grafts showed an absence of MID [25,27]. MFC targeting leukemia-associated immunophenotype (LAIP) did not detect MID in five patients who received chemotherapy prior to OTC [26]. Immunohistochemistry did not detect any markers in OT of seven patients, though for one of them RT-PCR was positive [37]. One childhood cancer survivor had her first autotransplantation in 2017 prior to receiving a negative result of *BCR-ALB* transcript in analyzed OT [36].

***AML***—MID was evaluated using IHC, RT-qPCR, PCR or MFC in OT before and after xenotransplantation in immunodeficient mice. Specific molecular markers identified MID in 3 out of 10 analyzed pediatric AML patients using RT-qPCR and MFC [25,26]. Two of them had started chemotherapy and one did not have treatment prior to OTC. *BCR-ABL1* fusion transcript was found in one patient by RT-qPCR in the OT before but not after long-term xenotransplantation (22 weeks) in immunodeficient mice [25]. MFC by LAIP resulted in positive MID in ovarian cortical tissue of 2/3 patients (detected LAIPs: CD13, CD33, CD117, CD65, CD7, HLA DR2, CD34, CD38 or CD13, CD33, CD65, CD117, CD11c) before but not after xenotransplantation (24 weeks) in immunodeficient mice [26]. For the remaining 7/10 patients, PCR, RT-qPCR, IHC or xenotransplantation did not show the presence of cancer cells in OT, though only 1 of them had initiated the treatment before OTC.

***CML***—MID was analyzed in the OTs of five CML patients in three studies (Table 2). All had their OTs cryopreserved prior to chemotherapy. Malignant cells were identified by *BCR-ABL* fusion transcript using quantitative RT-PCR in three out of five patients (60%) [35,37]. IHC failed to detect CML cells though RT-PCR was positive. RT-PCR did not identify infiltration of CML cells in OT prior or after xenotransplantation in the remaining two patients [27].

***JMML***—IHC staining of CD3, CD4 and CD68 revealed no MID in OT of one JMML patient [37]. The patient had no chemotherapy prior to OTC.

***Burkitt’s lymphoma***—RT-qPCR showed the absence of *IgH* rearrangements in one patient, though the patient had started chemotherapy [38].

### 3.3. Central Nervous System (CNS) Tumors

Two studies (Table 3) reported MID testing in OT of patients diagnosed with astrocytoma, ependymoma, germinoma, glioblastoma, medulloblastoma and PNET (primary CNS localization), with only one case being metastasized disease (a PNET patient). Ovarian tissues from 17 patients with a mean age at the time of cryopreservation of 8.2 ± 4.7 years (median 8 years, range 1–17 years) were analyzed. While OTC was performed for all 17 patients before chemotherapy, the slow freezing technique for OTC was reported only for 2 patients. All OTs analyzed were MID-negative by IHC or RT-ddPCR, with molecular markers including NSE and GFAP or *GFAP* and *ENO2*, or xenotransplantation [28,29]. However, we assume that patients described in [28] are included in a more recent article [29]. To avoid duplication of patients, only information on how the OTs were analyzed was included in Table 3 and Appendix A.

**Table 3 cancers-15-04199-t003:** Summary studies on minimal infiltrative disease in cryopreserved ovarian tissue of girls with central nervous system tumors.

Ref.	Study Type	Pt(*n*)	Type of Cancer	Markers and Methods (1–5)	Detected by Methods(1–5)	Positive Patients, *n* (%)	Potential Bias	Overall Bias
Histology (1)	IHC (2)	PCR ^$^ (3)	Xeno (4)	NGS (5)	SB	AB	MB	DB	SC	SA*	
[29]	Prospective study	17	Astrocytoma, ependymoma, germinoma, glioblastomsa, medulloblastoma, PNET	*n* = 17	NSE, GFAP*n* = 17	*GFAP*, *ENO2**n* = 14	*n* = 17	-	ND	-							
[28]	Case reports	3	PNET	*n* = 3	NSE, GFAP*n* = 3	*GFAP*, *ENO2**n* = 3	*n* = 3	*n* = 1	ND	-							
**Grade assessment**
Study design	Low	Prospective cohort study in 1/1 (case report is not incorporated in the body summary of evidence since the same patients are in the prospective study)
Study limitations	−1	Some limitations: SB moderate in 1/1; AB low in 1/1; MB is moderate in 1/1; DB is moderate in 1/2; SC moderate in 1/1; SA low in 1/1
Consistency	NA	Only one study
Directness	−1	Though results are direct, population and outcomes cannot be generalized
Precision	−1	Low number of patients; only one study
Publication bias	0	Unlikely
Effect size	0	NA
Dose–response	0	NA
Plausible confounding	+1	Treatment as plausible confounder may seriously influence the effect
**Quality of evidence**	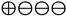	**Very low**
**Conclusion**		Detection of MID in ovarian tissues from central nervous system tumors patients is unlikely despite various attempts to identify it but the results cannot be generalized (1 study, 17 participants)


 = low risk of bias; 

 = moderate risk of bias; 

 = high risk of bias; 
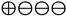
 = very low quality of evidence. AB = attrition bias (the study data available (i.e., participants not lost to follow up) adequately represent the study sample), DB = detection bias (the outcome of interest is measured in a similar way for all participants), FISH = fluorescence in situ hybridization, IHC = immunohistochemistry, L = low risk, M = moderate risk, MB = measurement bias (the prognostic factor is measured in a similar way for all participants), MID = minimally infiltrative disease, ND = not detected, OT = ovarian tissue, OTC = ovarian tissue cryopreservation, PCR = polymerase chain reaction, PNET = primitive neuroectodermal tumor, Pt = patients, SA* = no statistical analysis performed, all primary outcomes are reported, SB = selection bias (the study sample adequately represents the population of interest), SC = study confounding (important potential confounding factors are appropriately accounted for), Xeno = xenotransplantation (and evaluation after). ‘-’ means the method was not used in the specific study; ‘^$^’ includes all types of PCR: PCR, RT-PCR, quantitative RT-PCR, digital droplet RT-PCR, nested PCR.

## 4. Discussion

In 17 studies, 25 patients tested positive for MID in the OT of at least 115 pediatric patients diagnosed with solid tumors (Table 1), hematological malignancies (Table 2) and CNS tumors (Table 3). The most commonly used methods for detection of MID in OT were histology, IHC and RT-qPCR. For 62 patients, additionally, OTs were xenotransplanted into immunodeficient mice and reevaluated for MID [16,24,25,26,27,28,29]. 

A substantial proportion of the included articles focused on EWS, since EWS treatment harbors a high risk of infertility due to the high cyclophosphamide equivalent dose score of the protocols used. Three patients who were identified as being MID-positive had their OT frozen before starting treatment and two were reported to have a localized disease. All reported patients with metastatic disease (*n* = 11) were found to be negative for MID, while most of them had OTC performed before starting chemotherapy. This is a very important finding, indicating that the initial stage of the disease in EWS is not a major predictive value for MID’s presence in cryopreserved OT and suggests that the diagnosis of disseminated EWS may not preclude offering OTC to the patients. Evaluating possible infiltration in hematological malignancies is another challenge. Since leukemic cells transported with blood flow are supposed to be omnipresent, presumably there is a much higher chance to detect MID prior to the initiation of chemotherapy. On the other hand, leukemic cells are extremely sensitive to chemotherapy and may start disappearing after initiation of treatment [39]. However, we observed that 12 out of 22 MID-positive patients had received chemotherapy prior to OTC and 10 patients had not. In summary, 71 patients had their OTs cryopreserved before treatment and 41 after treatment initiation. Though the moment when the OT was obtained—prior to the initiation of chemotherapy or during the treatment—is important, this confirms that treatment status is not the only confounder in the detection of MID. 

Though CNS tumors have a 0.5–18% capacity to metastasize [29], to date only one case of a pediatric brain tumor with ovarian involvement has been reported [40] which makes metastasis relatively uncommon in children. In our systematic review of 17 CNS tumor patients, we identified that 100% of OT fragments were free of cancer cells, even in those who had experienced relapse or died during the treatment later, suggesting that CNS tumors may carry a low risk of ovarian infiltration [28], though larger-scale studies are needed.

The presence of malignant cells in OT is the subject of ongoing research and there are no widely established detection methods due to limited availability of ovarian tissue for analysis. Some general methods include conventional histology and IHC, however, their sensitivity and power are low with a detection limit of 1% or more tumor cells [41]. Targeted molecular-genetic approaches such as PCR or RT-qPCR with sensitivity of 10^−3^ to 10^−6^ seem to be better applicable and even broader and deeper screening methods such as RT-ddPCR or next generation sequencing with sensitivity of <10^−5^ to 10^−6^ are getting to the front lines in MID detection [42]. To achieve even higher sensitivity, sometimes methods are used in combination, however, it should be based on the disease context and sample availability while collaborating with experts to obtain the best detection power. Future studies to develop and validate the methods for MID are necessary to optimize methods and to enable required sensitivity. In addition, xenotransplantation of OT into immunodeficient animals may be used as a preclinical method to predict relapse [43]. However, xenotransplantation is a time-consuming method and the most important question is how results can be implemented in clinical care. Previous ex vivo experiments have shown that transplanted cryopreserved OT fragments in immunodeficient mice can vanish or decrease in size, thereby complicating analysis of MID [26,27,28]. It has also been reported that transplantation of a small number of cells may be insufficient to cause the disease, which means that the detected MID with highly sensitive methods might not have any clinical relevance, though a positive result should not be ignored [44,45]. Similar observations by other scientists were made when OTC was performed after induction of treatment or after achieving complete remission [45,46]. However, chemotherapy may not exclude malignant cells completely and the risk of them harboring within OT remains [26,27]. It has also been suggested that exploration by histology [24], blood blast count [25] or MCF and PCR-verified infiltrations into distant organs, such as bone marrow, spleen or lymph nodes [26], of xenotransplanted animals may be useful. MFC seems to be a very promising, adaptable technique for searching specific markers in leukemia patients, and it has already been used by some research groups on OT [47], testicular tissue [44], bone marrow or peripheral blood as a predictive tool of clinical outcome [48,49] with sensitivity of 10^−3^ to 10^−4^ [42].

This systematic review uncovered gaps of knowledge: no studies have been reported on ovarian involvement in children with renal tumors, hepatoblastomas, retinoblastomas or Hodgkin lymphomas (Table 4). These cancers carry specific molecular (driving) markers that can be used as candidates to identify MID [50]. In addition, the OTs of OS patients have not been tested using specific markers for MID detection or sensitive methods [24,33]. However, there was an attempt to identify molecular markers by FISH or RT-PCR in OS patients but without any success [34] and only later a valuable candidate RB1 as a prognostic factor was proposed [51]. Additionally, Dolmans et al. (2016) [34] claim that if no specific tumor markers are found, xenotransplantation into immunodeficient mice should be used for the detection of malignant cells before any reimplantation. The absence of specific markers at the time of diagnosis for some cancers underscores the importance of a comprehensive analysis of primary tumor tissues or finding new ways of detecting markers. Moreover, this systematic review indicated that the most used freezing technique for OTC was slow freezing. Only one study by Chaput et al., in 2019 [31], compared the impact on RNA yield after freezing OTs by slow freezing and snap freezing techniques. No differences between the two methods that could potentially have influenced MID detection by molecular methods were described/found.

### 4.1. Strengths and Limitations

To our knowledge, this is the first systematic review on MID in cryopreserved OT of pediatric oncology patients that includes published markers and detection methods used for screening. We summarized 17 selected articles that we considered to be representative of the current state of the field. However, due to variations in techniques for detection, MID markers, metastasis status (in cases of solid and CNS tumors) and time of OTC (prior, during, after the treatment) and low sample sizes (ranging from 1 to 17 patients) in the studies, a meta-analysis was not feasible.

### 4.2. Implications for Clinical Practice and Research

Two recent reviews described the current status of strategies to safely use cryopreserved OT for fertility restoration in adult [53,54] and prepubertal (<12 years) [54] oncological patients, including in vitro maturation, construction of an artificial ovary, maturation of oocytes, stem cell-based oogenesis and purging. The reviews conclude that in pediatric oncology these methods are all still experimental procedures and require substantial additional research before clinical application. As OTC and subsequent OT transplantation remain the only option for fertility restoration for girls with cancer, this highlights the need for precise MID detection in OT. While encouraging results have been published for young girls with sarcomas and CNS malignancies, we can never exclude the possibility of the harboring of malignant cells within ovarian tissue. Table 5 shows proposed and potential markers and techniques to detect MID according to the markers used in ovarian tissue but also those used for disease detection and/or those of prognostic value. Although some of the proposed and potential markers are usually found in the original tumor tissue and less likely to be found in MID due to the small amount of DNA, with the newest methods such as NGS, even the smallest alterations could be found after careful accuracy and reliability validation. NGS panels after identification of disease-specific targets may be the best approach for MID detection. In this way, it would be possible to reach deeper sequencing coverage, use data streamlining and provide the best personalized medicine approach to the patients [55]. 

## 5. Conclusions

In conclusion, testing OTC harvests for MID is important for individual patients. If a patient expresses the desire for autotransplantation of harvested OT, it involves a preliminary MID-based risk assessment depending on their cancer type and molecular markers found at the diagnosis by an expert team. Applying the most reliable methods, as was shown in this systematic review, such as RT-qPCR, FISH or MFC, for determining the absence of MID, or alternatively, in the case of existing MID, purifying the OT from malignant cells to ensure oncologically safe future fertility restoration, is important for the clinical implementation of autotransplantation of cryopreserved OT material in girls with cancer.

## Figures and Tables

**Figure 1 cancers-15-04199-f001:**
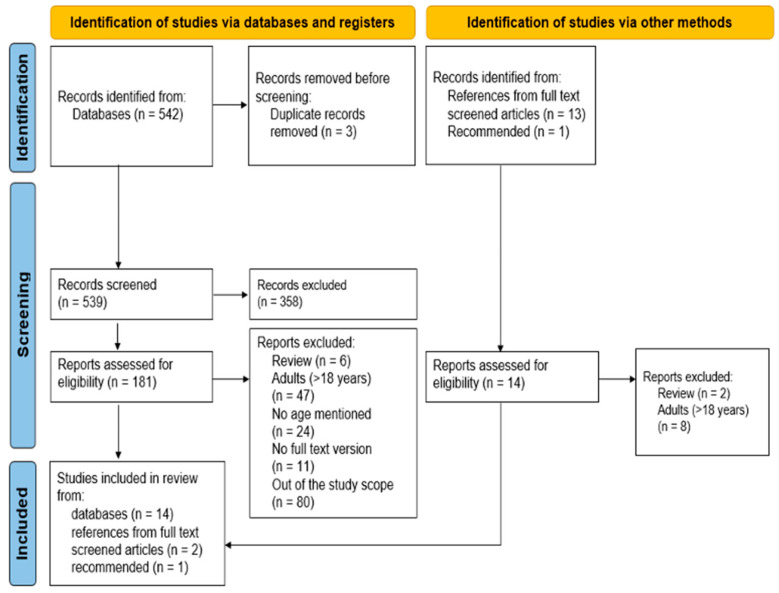
Flow diagram of the study selection process. The total number of full-text articles screened was 195; 17 of them were included in the systematic review and 178 were excluded. Assembled according to PRISMA guidelines [21].

**Table 4 cancers-15-04199-t004:** Reported MID in harvested ovarian tissue involvement in cancer patients. MID—minimal infiltrative disease.

Incidence Rate	Types of Cancer in Adults	Types of Cancer in Children
High risk >11%	Leukemia, Burkitt’s lymphoma, neuroblastoma	Ewing sarcoma, acute lymphoblastic leukemia, acute myeloid leukemia, chronic myeloid leukemia
Moderate risk 0.2–11%	Cervical adenocarcinoma, breast cancer (infiltrating lobular subtype, stage IV), Ewing’s sarcoma, non-Hodgkin lymphoma, colon cancer	
Low risk <0.2%	Remaining malignancies	Osteosarcoma, rhabdomyosarcoma, synovial sarcoma, clear cell sarcoma, Burkitt’s lymphoma, juvenile myelomonocytic leukemia, central nervous system tumors

Classification of MID incidence in adults is from [52]; suggestion of classification of MID incidence rates in pediatric patients is according to the findings of this systematic review.

**Table 5 cancers-15-04199-t005:** Proposed and potential markers with techniques to detect minimal infiltrative disease in cryopreserved ovarian tissue of girls with specific malignancies.

Malignancy	Proposed Markers	Potential Techniques	References
Overall	***ABL* class abnormalities**	**RT-qPCR**	[25,27]
	*NUP98* gene fusion transcripts	RT-PCR	[56]
	*WT1* mutations	PCR, qPCR	[57]
	*RASSF1A* hypermethylation	Quantitative MSP	[58]
	*P53* mutations, transcript levels	NGS, RNA sequencing	[59,60]
	*Notch* aberrations	Sequencing	[61]
	CD19, CD20, CD22	MFC, NGS, RT-qPCR	[62]
	GD2, B7H3	MFC	[63]
Acute lymphoblastic leukemia	**Philadelphia chromosome-positive (Ph^+^; *BCR-ABL1* fusion gene)**	**PCR, RT-PCR**	[25,36]
	***(TCF3) E2A* fusion genes**	**RT-qPCR;** **RT-ddPCR**	[16,38]
	**CD34, CD10, CD20, CD79a, CD3, TdT**	**IHC**	[37]
	**CD19, CD34, CD10-, negative for myeloid markers, CD45, HLA-DR2, CD10, CD19, CD22, CD33, CD38, CD2, cyCD3, CD5, CD7, CD33, CD45RA, CD123**	**MFC**	[26]
	***Ig(H/K)/TCR* targets**	**RT-PCR, RT-ddPCR**	[16,25,27,38]
	** *RUNX1 (AML1)* ** **translocations, fusion genes**	**RT-PCR, FISH, RT-qPCR,** PCR	[25,37,38,64]
	***MLL (KMT2A)* rearrangements,** patient-specific on DNA	**RT-qPCR,** long-distance inverse PCR	[38,65]
	Intrachromosomal amplification of chromosome 21 (iAMP21)	FISH	[64]
	*IKZF1* mutations	Sequencing	[66]
	*SIL-TAL1* fusion gene	PCR	[67]
	Asparagine synthetase	RT-qPCR	[68]
Chronic myeloid leukemia	***BCR-ABL* fusion gene**	**FISH, RT-PCR,** PCR	[27,37,69]
	**MPO, glycophorin A, CD34, CD68, CD117, LCA/CD45, Factor VIII**	**IHC**	[35,37]
Acute promyelocytic leukemia	*PML-RAR*α brc isoforms	RT-qPCR, FISH	[70]
Acute myeloid leukemia	***RUNX1 (AML1)* fusion genes**	**RT-PCR, RT-qPCR**	[38,71,72]
	***NPM1* mutations**	**RT-qPCR**	[38]
	** *FLT3* ** **fusion transcripts**	**RT-PCR**	[25]
	***BCR-ABL1* fusion genes**	**RT-PCR**	[25]
	**CD34, CD68, CD117, MPO, CD13, CD4**	**IHC**	[37]
	**CD34, CD33, CD13, CD117, CD38, CD65, CD7, HLA-DR2, CD11c**	**MFC**	[26]
	*CBFB-MYH11* fusion genes	**PCR**, RT-qPCR	[37,71]
	*NUP98* rearrangements	RT-qPCR, NGS	[56,73]
	*FUS-ERG* fusion gene	FISH, PCR, RNA sequencing	[74]
	*KMT2A* rearrangements	RT-qPCR, FISH, RT-PCR; long-distance inverse PCR	[65,75]
	*RBM15-MKL1 (OTT-MAL)* fusion gene/transcript	RT-PCR	[76]
	*CEBPA* mutations	PCR, sequencing	[77]
	*UBTF* mutations	NGS	[78]
Acute myeloid leukemia–Down syndrome	*GATA1* mutations, patient-specific on DNA	Sequencing	[79,80]
Anaplastic large cell lymphoma	*ALK* fusion genes, *NPM1* mutations	RT-PCR, FISH, immunofluorescence	[78,81]
Juvenile myelomonocytic leukemia	**CD3, CD4, CD68**	**IHC**	[37]
Lymphoma (Hodgkin and non-Hodgkin)	**Immunoglobulin (Ig) or T-cell receptor (*TCR*) gene rearrangements**	**RT-qPCR**	[38]
		Multiplex PCR	[82]
	9p24 amplification/*JAK2*	DNA copy number analysis, RT-qPCR	[83]
	14q11/*TRA/D*	FISH	[84]
Wilms tumor	*WT1*, *WTX* mutations	PCR, qPCR	[57]
	*CTNNB1* mutations	qPCR	[57]
	*DROSHA/DGCR8* mutations	Sequencing (DNA, RNA)	[85]
	*SIX1/SIX2* mutations	Sequencing (DNA, RNA)	[85]
	*DICER1* mutations	Sequencing (DNA, RNA)	[85]
	*DIS3L2* mutations	Sequencing (DNA, RNA)	[85]
	*FBXW7* mutations	Chromosome copy number profiling, sequencing	[86]
	*DIS3L2* mutations	PCR	[87]
	*TP53* mutations	NGS	[88]
Clear cell sarcoma of the kidney	*BCOR* internal tandem duplications	RNA sequencing, PCR, IHC, FISH	[88,89]
	*YWHAE-NUTM2* fusion transcript	RT-PCR	[90]
	*TCF21* hypermethylation	Methylation-based methods, quantitative pyrosequencing methylation analysis	[90,91]
Malignant rhabdoid tumor of the kidney	*SMARCA4*; *SMARCB1*	IHC, WGS	[92,93]
Renal cell carcinoma	*TFE3* (Xp11) translocations/fusion transcripts	FISH, WGS/NGS, IHC, RNA sequencing	[60,88,94,95]
	*TFEB* translocations/fusion transcripts	RNA sequencing, FISH, IHC	[60,88,95]
	*ALK* rearrangements	IHC, FISH, RT-PCR, NGS	[96]
Neuroblastoma	*MYCN* amplification/mutations	FISH, ddPCR, NGS	[88,97]
	PHOX2B	IHC	[98]
	*TH*	RT-qPCR, multiplex RT-qPCR	[99]
	*CHRNA3*, *DBH*, *GAP43*, *POSTN*, *PRRX1* and *FMO3*	multiplex RT-qPCR	[100]
	*MDM2*	RT-qPCR	[101]
	*ATRX* mutations	WGS, IHC	[102]
	*ALK* mutations	Sequencing, qPCR	[103]
Synovial sarcoma	**Bcl-2, *SYT-SSX* fusion gene**	**IHC, RT-qPCR**	[34]
Clear cell sarcoma	**Melanoma cocktail/S100**	**IHC**	[34]
	***EWS-ATF1* fusion gene**	**RT-qPCR**	[34]
	*BROC* mutations	RT-qPCR	[104]
	*YWHAE-NUTM2* fusion transcript	RT-qPCR	[104]
Ewing Sarcoma	***EWS* fusion transcripts**	**RT-qPCR**	[15,24,30,31,32,33,34]
	**t(X;22) *EWS* translocations**	**FISH**	[34,88]
	**CD99, INI1**	**IHC**	[34,88]
Osteosarcoma	*RB1* deletion	IHC, PCR, RT-PCR	[51]
Rhabdomyosarcoma	***PAX3/7-FOXO1* fusion gene**	**RT-qPCR**	[34,105]
	** *MYOD1* **	**IHC**, RT-qPCR	[34,105]
	** *MYOGENIN* **	**IHC**, RT-qPCR	[34,105]
	*VGLL2* fusion transcripts	RT-PCR, RNA sequencing	[106]
	*NTRK* fusion transcripts	RT-PCR, RNA sequencing	[106]
	*(B)RAF* fusion transcripts	RT-PCR, RNA sequencing	[106]
	t(2;13) translocation alveolar	RT-PCR	[107]
	cfRRBS	shWGS	[108]
Medulloblastoma	**GFAP/NSE**	**IHC**	[29]
	Alterations in WNT and SHH pathways’ components	FISH, sequencing, methylation-based methods and combinations of them	[109]
Astrocytoma	**CD99/NSE** ** *GFAP* **	**IHC, RT-ddPCR**	[29]
Ependymoma	**GFAP/NSE** ** *GFAP* **	**IHC, RT-ddPCR**	[29]
Germinoma	**GFAP/NSE** ** *GFAP* **	**IHC, RT-ddPCR**	[29]
Glioblastoma	**GFAP/NSE** ** *GFAP* **	**IHC, RT-ddPCR**	[29]
Primitive neuroectodermal tumor	**GFAP/NSE** ** *GFAP* **	**IHC, RT-ddPCR**	[29]
Germ cell tumors	Chromosome 3p gain, miRNA	NGS, RT-qPCR	[110]

The markers and techniques in **bold** have been reported for testing in ovarian tissue (results from this review), the markers and techniques not in bold have not yet been described for detection in ovarian tissue but are used in the diagnostic setting. ddPCR—digital droplet PCR, FISH—fluorescence in situ hybridization, IHC—immunohistochemistry, MFC—multicolor flow cytometry, MSP—methylation-specific PCR, NGS—next generation sequencing, PCR—polymerase chain reaction, qPCR—quantitative PCR, RNA—ribonucleic acid, RT-PCR—real-time PCR, RT-qPCR—real-time quantitative PCR, shWGS—shallow WGS, SVS—synovial sarcoma, WGS—whole genome sequencing.

## Data Availability

All data are presented in the manuscript and Appendix A.

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
