# Peer review of "Minimal Infiltrative Disease Identification in Cryopreserved Ovarian Tissue of Girls with Cancer for Future Use: A Systematic Review"

_cancers, 2023, doi:10.3390/cancers15174199_

Round 1
Reviewer 1 Report
Grubliauskaite et al. report on the minimal infiltrative disease (MID) in cryopreserved ovarian tissue (OT) of girls with cancer. The review is of interest and covers an important topic. The selection of articles is relevant and offers a comprehensive view of current detection methods and markers to study MID in ovarian tissue and the clinical implications of this investigation.
However, I have some comments.
As stated at the end of the Introduction, the authors focus on tumor markers and their detection methods to identify MID in OT. Currently used approaches are well described, but the author’s choice of potential markers and methods needs to be better explained and discussed.
Table 5 contains a great deal of information and some prioritization/adjustment is needed. For the reader, it would be helpful to highlight the established approaches for MID detection to distinguish them from markers and methods that have not yet been validated. Some of the proposed markers have diagnostic and/or prognostic values but they are hardly applicable to MID detection in OT. What were the criteria for the selection of markers and methods? Such events as DNA hypermethylation, duplications, gains, deletions and alterations in pathways can be detected in primary tumors, but they are undetectable when tumor cell fraction is low and that is the case of MID. Similarly, the methods as Sanger sequencing, Southern blot, SNP array, telomere FISH and karyotyping have low sensitivity. In addition, karyotyping requires cells culture to obtain metaphases.
In the third paragraph of the Discussion, the methods to detect malignant cells are briefly listed, but there is no comment on their specificity, sensitivity and performance. Only multicolor flow cytometry is mentioned as a promising and adaptable technique. What is the place of other methods? How to combine methods to obtain the best detection power? What is the best strategy for using next generation sequencing for this application? How NGS could be validated and what is the perspective of its introduction in clinical routine for MID detection?
Minor: In Table 4, low risk should be “<0.2 %”.
Author Response
We appreciate the extensive review of our manuscript and the valuable suggestions how to enhance the content. Please find our point-by-point response to the comments.
Point 1: Table 5 contains a great deal of information and some prioritization/adjustment is needed. For the reader, it would be helpful to highlight the established approaches for MID detection to distinguish them from markers and methods that have not yet been validated.
Response 1: We appreciate the opportunity to address the comment related to selection of potential methods and markers. Firstly, we would like to start by highlighting that there are no validated approaches for detection of minimal infiltrative disease (MID) in ovarian tissue yet, due to limited availability of ovarian tissue for analysis. We agree this was not clearly specified in the manuscript and therefore added text in the 4th paragraph of the Discussion section. In addition, in Table 5 we have bolded the methods and markers now which have been used for MID detection, in the identified papers, according to this systematic review. We added the following in the legend of the table to explain this:
“The markers and techniques in bold have been reported for testing in ovarian tissue (results from this review), the markers and techniques not in bold have not yet been described for detection in ovarian tissue but are used in the diagnostic setting.”
Furthermore, we amended the order of the table to put the markers tested in ovarian tissue first to increase applicability of the table. By doing so, it may be easier for the reader to distinguish between proposed, yet not published markers/methods and the ones that have been used before by other scientists. The not published methods are derived from standard methods used for MRD (minimal residual disease) detection in childhood leukemia settings, for which we include the references.
Point 2: Some of the proposed markers have diagnostic and/or prognostic values but they are hardly applicable to MID detection in OT. What were the criteria for the selection of markers and methods? Such events as DNA hypermethylation, duplications, gains, deletions and alterations in pathways can be detected in primary tumors, but they are undetectable when tumor cell fraction is low and that is the case of MID. Similarly, the methods as Sanger sequencing, Southern blot, SNP array, telomere FISH and karyotyping have low sensitivity. In addition, karyotyping requires cells culture to obtain metaphases.
Response 2: We agree with Reviewer that structural aberrations such as duplications, gains, deletions as well as mutations can be undetectable when tumor cell fraction is low and that thresholds need to be developed for reliable MID testing but that is beyond the scope of this review. Designing selection and international validated and standardized methods will be the aim of future studies. We here summarized available information on potential MID markers from the experience of molecular characteristics in pediatric cancer, and MRD studies, that have been published. This is according to the development of MRD detection in leukemia that has been accomplished over the past decades.
We have also added the following text to 4.2. section:
“Table 5 shows proposed and potential markers and techniques to detect MID according to the markers used in ovarian tissue but also those used for disease detection and/or prognostic values. Although, some of the proposed and potential markers are usually found in the original tumor tissue and less likely to be found in MID due to small amount of DNA, with the newest methods such as NGS, even the smallest alterations could be found after careful accuracy and reliability validation.“
The criteria for the selection of markers were the possibility to use the molecular characteristics of the cancer cells, which would not be expected to be found in normal tissue cells for detecting MID.
Point 3: In the third paragraph of the Discussion, the methods to detect malignant cells are briefly listed, but there is no comment on their specificity, sensitivity and performance. Only multicolor flow cytometry is mentioned as a promising and adaptable technique. How to combine methods to obtain the best detection power?
Response 3: We agree this is an important question and addresses the future options for further designing methods, validation and implementation. For now, with the available evidence, we can just summarize the information of the series published which are presented in the manuscript.
To clarify this, we have added this summarized information about specificity, sensitivity and performance of methods in published series in the 4th paragraph of Discussion section of the revised version of the manuscript:
“Some general methods include conventional histology and IHC, however, their sensitivity and power are low with detection limit of 1% or more tumor cells [41]. Targeted molecular-genetic approaches such as PCR or RT-qPCR with sensitivity of 10-3 to 10-6 seem to be better applicable or even broader and deeper screening methods such as RT-ddPCR or next generation sequencing with sensitivity of < 10-5 to 10-6 are getting to the front lines in MID detection [42]. To achieve even higher sensitivity sometimes methods are used in combination, however, it should be based on the disease context and sample availability while collaborating with experts to obtain the best detection power. Future studies to develop and validate the methods for MID are nescessary, to optimize methods and to enable required sensitivity.”
Point 4: What is the best strategy for using next generation sequencing for this application? How NGS could be validated and what is the perspective of its introduction in clinical routine for MID detection?
Response 4: We appreciate this interesting question from Reviewer. We feel that NGS method can be used as long as it enables to detect the patient-cancer specific targets that are specific for an individual patient and detection limits can be standardized across centers and countries. Hence, the resources in individual hospitals will need to lead that decision.
Added text in the 4.2. section of the revised manuscript:
“NGS panels after identification of disease-specific targets, may be the best approach. In this way it would be possible to reach deeper sequencing coverage, use data streamlining and provide best personalized medicine approach to the patients [55]”.
Future studies needed to standardize and validate the best strategies of implementation of NGS in the clinical routine for MID detection.
Point 5: Minor: In Table 4, low risk should be “<0.2 %”.
Response 5: Thank You for noticing, we have corrected it.
Reviewer 2 Report
The entitled manuscript „ Minimal infiltrative disease identification in cryopreserved ovarian tissue of girls with cancer for future use – a systematic review” by Monika Grubliauskaite et al. was aimed to write a systematic rewiew article on minimal infiltrative disease (MID) within ovarian tissue. The article contains 1 figure and 5 tables. In my opinion, the many tables make it difficult to read. There could be a summary diagram at the end.
Author Response
We appreciate the reviewing of our manuscript and we are deeply thankful for the comment.
Point 1: The article contains 1 figure and 5 tables. In my opinion, the many tables make it difficult to read. There could be a summary diagram at the end.
Response 1: Though we do agree with Reviewer that where is a significant amount of information provided and could be difficult to read at first, we believe it is the best way to present the limited information on this topic and make it a valuable tool for future updates and analysis for MID detection in ovarian tissue of pediatric patients. As well as we split the information to three (per tumor type) tables in order to report the most related information together and easier to find to a specific oncologist/scientist.
Reviewer 3 Report
Comments to the authors:
In this systematic review entitled “Minimal infiltrative disease identification in cryopreserved ovarian tissue of girls with cancer for future use – a systematic review,” the author(s) have picked up an important and relevant topic. It is interesting as it is focused on MID in pediatric oncology patients and provides good up-to-date information from the relevant literature. Although the number of selected articles (17) related to the topic is very less and the data presented from the number of girls is also less (115), still authors have done a commendable job in presenting the limited available information in a nice way. This will help in better understanding the missing information now and thinking about the future possible options while preserving the OT and the test required at the time of transplantation.
This review being part of a project regarding childhood cancer care and focussing on the targets used to screen MID and techniques involved in MID detection in OT might serve as a base for future planning and strategies related to better screening and fertility options.
The authors have written it well discussing the search strategy, selection criteria, data extraction, and quality of evidence assessment. Results are presented sequentially with adequate information in each section and with proper reference. The summarized information in the form of a table for all three categories (solid tumors, hematological malignancies, and CNS tumors) that the authors have discussed in this review will make it convenient for general readers to understand in a short time.
The English language is fine. Authors should go through the manuscript to remove a few grammatical errors.
The authors should add some details about the age/time when OT was preserved and also the methods of cryopreservation for the selected patients, and do these methods have some role to play in the screening of MID and the success of transplantation?
The English language is fine. Authors should go through the manuscript to remove a few grammatical errors.
Author Response
We appreciate the extensive review of our manuscript and the valuable suggestions how to enhance the content. Please find our point-by-point response to the comments.
Point 1: The authors should add some details about the age and time when OT was preserved.
Response 1: We agree with Reviewer that the age of the patients who underwent ovarian tissue cryopreservation can be reported more clearly. We have now specified this information in the first paragraph of each specific section (e.g., 3.1., 3.2., 3.3.) in the revised version of the manuscript. As the focus of the review is on pediatric patients (< 18 years), we hope that the provided information is now sufficient to address the age-related aspects.
As discussed in the second paragraph of the Discussion section, the presence of minimal infiltrative disease (MID) can be identified regardless of whether ovarian tissue cryopreservation occurred before or after the initiation of treatment. However, naturally when initial treatment has been given before OTC, it may be possible that no more cancer cells will be found. This initial treatment was therefore considered a confounding factor. We have added additional information on the time of ovarian tissue cryopreservation for all patients if it has been reported in the included articles. Information can be found in the first paragraph of each specific section (e.g., 3.1., 3.2., 3.3.) and in Discussion section as summary:
“In summary, 71 patients had their OTs cryopreserved before treatment and 41 after treatment initiation”.
Point 2: The authors should add some details about the methods of cryopreservation for the selected patients.
Response 2: We understand the importance of discussing the methods of cryopreservation and subsequent success of ovarian tissue transplantation. Though it was not within the scope of this systematic review to delve into these specific topics, we agree with the reviewer that additional information about the freezing techniques for specific patients may be of value. We now added a summary of the used methods in the second paragraph of Results sections of the revised version of the manuscript as follows:
“The main cryopreservation technique used was slow freezing (used for 87 patients, 5 of them had additionally snap frozen OT pieces, while for the others used technique was not reported.“
and we specified the first paragraph of each specific section (e.g., 3.1., 3.2., 3.3.) whether the information was provided in the included articles.
Point 3: Do these methods have some role to play in the screening of MID and the success of transplantation?
Response 3: Upon the question of Reviewer, whether freezing influences success, we added information in Discussion section of the revised version of the manuscript as follows:
“Moreover, this systematic review indicated that the most used freezing technique for OTC was slow freezing. Only one study by Chaput et al., 2019 [25] compared impact after freezing OTs by slow freezing with that of snap freezing techniques on RNA yield. No differences between the two methods that could potentially have influenced MID detection by molecular method were described/found.”.
We would like to emphasize, that the number of reported cases involving reimplantation of ovarian tissues from pediatric patients is still limited. Therefore, we feel we should refrain from providing any conclusions regarding success of autotransplantation on this topic.
Please find the revised version of the manuscript resubmitted online.
Round 2
Reviewer 1 Report
The changes made have greatly improved the manuscript. The paper can now be accepted in its present form.